# Compact Quad-Port MIMO Antenna with Ultra-Wideband and High Isolation

**Zhengrui He * and Jie Jin**

School of Electrical and Information Engineering, Tianjin University, Tianjin 300000, China
* Correspondence: zhengrui_he@tju.edu.cn

**Abstract:** In this paper, we propose a compact and highly isolated four-port ultra-wideband MIMO antenna. The antenna elements achieve broadband characteristics by etching a metamaterial structure on the radiating patch and stepping the coplanar waveguide feed. The test results show that the unit antenna can operate from 1.8 GHz to 16.38 GHz with an absolute bandwidth of 14.58 GHz and a relative bandwidth of 160.4% with good radiation properties and gain. After that, a compact four-cell ultra-wideband MIMO antenna is designed by using polarization diversity technology with an overall size of 51.2 mm × 51.2 mm × 1.524 mm. The MIMO antenna can operate from 1 GHz to 17 GHz with an absolute bandwidth of 16 GHz and a relative bandwidth of 177.78%. To reduce the coupling between cells, four angled slits are etched on the common ground to improve the isolation of the MIMO antennas to 27.8 dB. The performance parameters of the proposed MIMO antennas are further validated through simulation analysis and measurements. Moreover, the diversity properties of MIMO antennas are analyzed in detail, demonstrating the applicability of the proposed antennas in UWB communication systems, which can also be used for satellite mobile communications and satellite fixed communication services.

**Keywords:** UWB; MIMO antenna; high isolation; compact; polarization diversity





## 1. Introduction

With the development of modern wireless communication systems, high data transmission rates and large channel capacities have become hot topics of interest for researchers. The two bottlenecks limiting the increase in data transmission rate are the limited available spectrum resources and the multipath fading of the channel, while ultra-wideband technology (UWB) [1] shows strong resistance to multipath fading and high-speed transmission capability, while having the advantages of strong penetration capability, low power spectrum density, accurate positioning, etc., and is regarded as the optimal information carrier for short-range transmission and high-speed wireless connectivity. Multi-input multiple-output (MIMO) technology can be well realized with large channel capacity and high spectrum utilization. Without increasing the transmitting power and occupying the channel bandwidth, MIMO technology achieves a significant increase in channel capacity and high transmission rate by placing multiple antennas at the transmitter and receiver simultaneously and using spatial diversity technology and the multipath transmission effect of the channel to make the channel capacity of the system proportional to the number of antennas.

UWB MIMO technology [2], on the other hand, combines UWB technology with MIMO technology to take advantage of both, achieving large channel capacity and high data transmission rate while avoiding multipath fading effects. UWB MIMO antennas have also become a research hot spot for scholars in recent years. The most important concern for UWB MIMO antennas is the low coupling of the antennas. With the rapid development of modern mobile communications, mobile terminals and wireless devices are becoming increasingly miniaturized and integrated. Along with the miniaturization of UWB MIMO

antennas, the physical distance between antenna elements will become closer and the degree of mutual coupling will rise, leading to stronger and stronger electromagnetic interference that will cause the antenna to fail to function properly. Therefore, how to improve inter-cell isolation is a central issue in UWB MIMO antenna research.

Common approaches of MIMO antenna to improve isolation include designing decoupling structures [3,4], opening slits [5], loading metamaterials, frequency selective surfaces [6], electromagnetic bandgap structures [7–9], introducing neutralization lines [10–13] and parasitic cell decoupling [14], as well as using defective ground structures [15,16] and polarization diversity techniques [17–19]. The two-port MIMO antenna in [3] uses a "T"-shaped decoupling structure between cells and successfully increases the isolation to more than 24 dB, but the bandwidth is only 2.33–6.71 GHz and the size is too large. The two ports MIMO antenna in [5] work at 3.1–10.6 GHz and achieves good isolation of 31 dB by etching the center slot on the ground plane, but the antenna size is 47 mm × 93 mm × 1.6 mm. In the MIMO antenna in [10], through the operation of polarization diversity and docking floor, the average isolation degree of the antenna in the working range is greater than 26 dB, and the working bandwidth is extended to 2.1 GHz to 20 GHz, but the ECC of the antenna is low and the size is too large. The four-port MIMO antenna in [12] has a compact structure and can operate from 3–13.5 GHz. The antenna adopts an orthogonal layout structure, but the isolation degree is very low and the ECC only meets <0.4. In summary, the antennas in the aforementioned references suffer from complex structures, limited isolation effects, and large sizes.

In this paper, we design a compact four-port MIMO antenna with high isolation and ultra-wideband. By changing the antenna feeding method and introducing "T" shaped metamaterial, the impedance bandwidth is extended and a UWB MIMO antenna covering 1–17 GHz is designed. The decoupling approach, which combines polarization diversity techniques with the introduction of a decoupling gap, has been proposed for the design of UWB MIMO antennas and can improve the isolation to more than 27.8 dB per port. The validity of this decoupling approach has been validated by simulation analysis and measurements. With stable gain and good diversity properties, the proposed MIMO antenna can be widely adopted in UWB communication systems and also used for satellite mobile communications, fixed communications and live television.

The main content of this paper is organized as follows: In Section 2, an antenna capable of coverage operation in the ultra-wideband regime is designed by principle analysis and simulation optimization, and the design principle, evolution process, critical parameters and results of far-field simulations of the antenna are analyzed and discussed. Section 3 is based on the design and analysis of a compact four-cell MIMO antenna based on UWB antenna elements and achieves low coupling and miniaturization through polarization diversity and the introduction of a decoupling gap. Section 4 is the test and verification work for the broadband characteristics and low coupling characteristics of the four-cell MIMO antenna, which mainly carries out the test work of S-parameters and far-field radiation pattern, and analyzes and illustrates the diversity characteristics of the MIMO antenna, including: ECC, DG, MEG, TRAC, etc.

## 2. Design of UWB Antenna Unit

Traditional microstrip antennas typically come with drawbacks such as narrow frequency bands and parameter variations depending on the dielectric substrate and are therefore generally less used in UWB antenna designs. A planar monopole antenna can perfectly avoid these drawbacks, with the advantage of a wide working band and insensitive parameter variation, and with the advantages that come with a regular microstrip antenna: low profile, small size, lightweight, low cost, easy integration, etc. Therefore, the design work in this paper starts with a rectangular monopole antenna. In this section, the UWB monopole antenna is analyzed and discussed from design principles, structural evolution, critical parameter analysis, and simulation analysis.

## 2.1. Antenna Structure Design

As UWB communications evolve and progress, ultra-wideband antennas should not only achieve miniaturization and low profile, but also take into account the simple structure and ease of integration required for processing design, as well as the more stringent requirements on impedance bandwidth. The main indexes of antenna design in this section are: working band covers 3.1–10.6 GHz, meeting VSWR < 2, realizing omnidirectional radiation and stable radiation characteristics. For the monopole antenna whose radiation patch is round, rectangular, triangular and other regular shapes, the cylinder approximation method can be used to estimate the minimum resonant frequency to meet the target requirements [20,21], so it is enough to adjust the size of radiation patch and ground appropriately.

Table 1a shows the model diagram of the UWB monopole antenna unit, where the red part is the rectangular radiating patch embedded with a metamaterial array, and the blue part is the surrounding ground structure. The antenna sample in Figure 1b is a single-sided printed antenna structure on a Teflon circuit board with an overall size of 25.63 mm × 21.8 mm × 1.524 mm, which is comparable to the size of One yuan coin, and the specific geometric parameters are shown in Table 1.

**Table 1.** Geometrical parameters of the UWB monopole antenna.

| Parameters | L | W | l | w | $W_1$ | gw | $fw_1$ | $fw_2$ | gl | $fl_1$ | $fl_2$ | m |
|---|---|---|---|---|---|---|---|---|---|---|---|---|
| Value/mm | 21.8 | 25.63 | 9.94 | 10.1 | 13.32 | 4.81 | 5 | 1.4 | 10.5 | 0.6 | 3.5 | 1.45 |

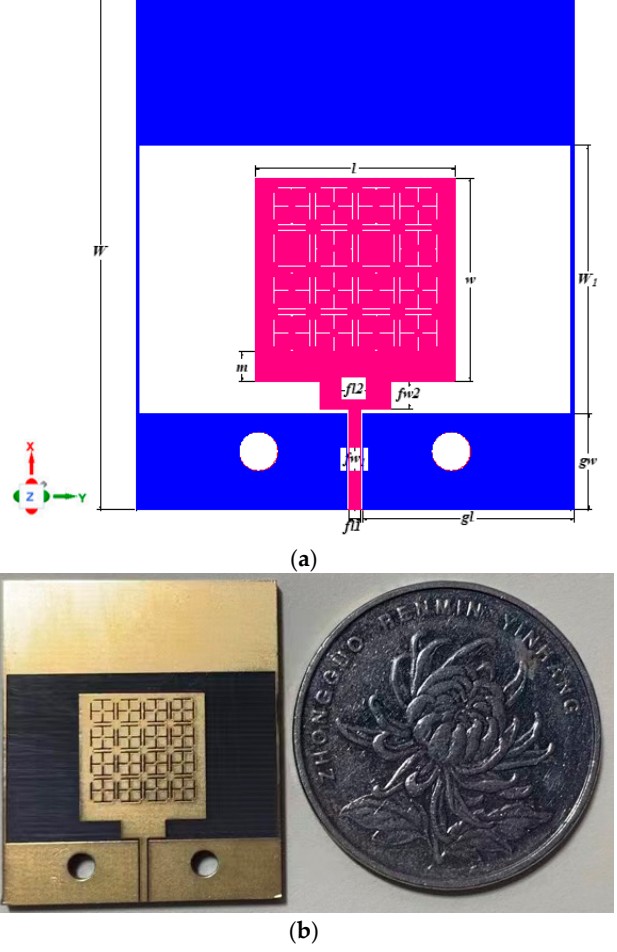

(a)

(b)

**Figure 1.** UWB monopole antenna: (**a**) structural diagram; (**b**) sample.

### 2.2. Design Procedure

To achieve the UWB communication band (3.1–10.6 GHz) specified by FCC, the basic monopole antenna is designed for improvement, and undergoes five steps in total, as shown in Figures 2 and 3. The rectangular monopole antenna in Figure 2 is the initial reference antenna, named Ref. Antenna, and the dielectric substrate is made of Teflon (PTFE) with a thickness of 1.524 mm, dielectric constant $\varepsilon_r = 2.1$ and loss tangent angle $\tan \delta = 0.0002$.

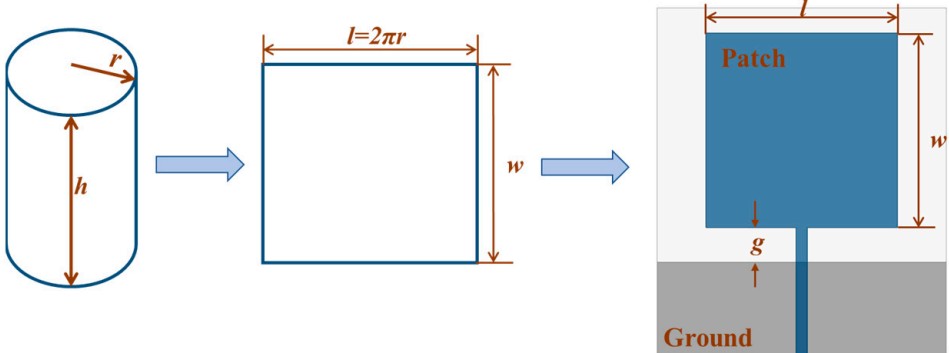

**Figure 2.** Rectangular patch planar monopole antenna estimated using the cylindrical approximation.

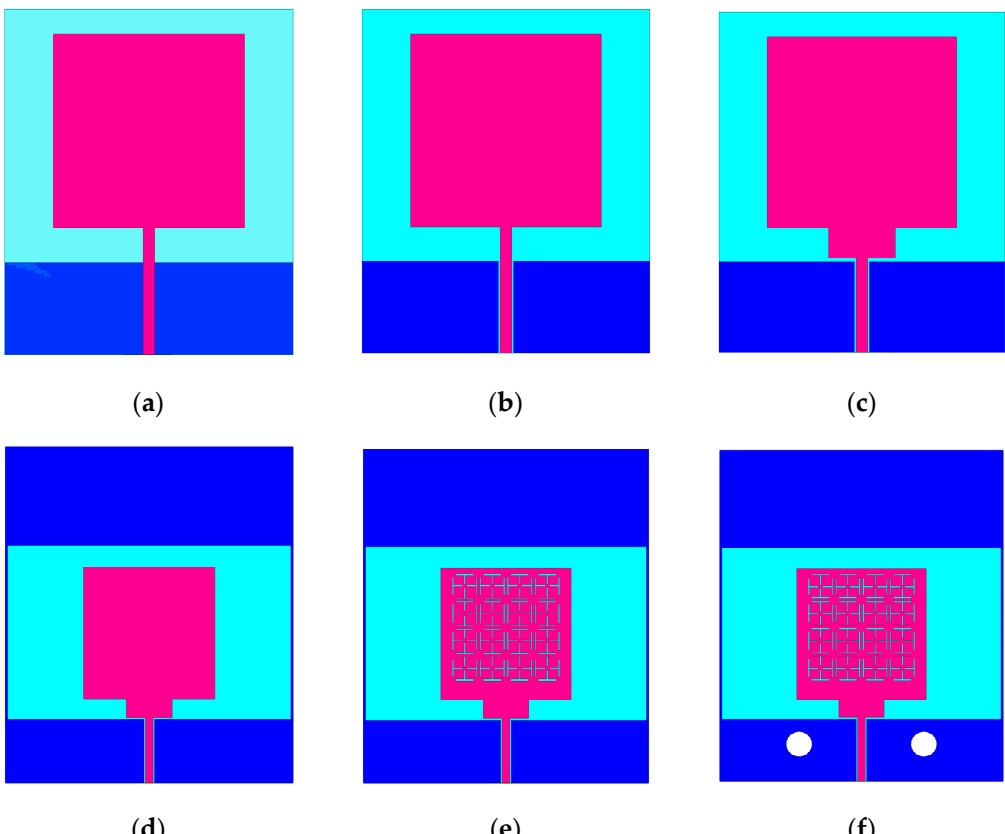

**Figure 3.** Design steps of UWB antenna: (**a**) Ref. Antenna; (**b**) first step; (**c**) second step; (**d**) third step; (**e**) Pro. Antenna; (**f**) perforated antenna model.

Using the cylindrical approximation method [20], the approximate dimensions of the rectangular radiating patch ($l \times w$) and the rectangular ground plane ($l' \times w'$) of Ref. Antenna can be calculated as follows:

$$l = 0.24 \times \lambda \times s \tag{1}$$

$$s = \frac{h/r}{1+h/r} = \frac{h}{h+r} = \frac{\omega}{\omega + l/2\pi} = \frac{2\pi\omega}{2\pi\omega + l} \tag{2}$$

Combined with Formulas (1) and (2), consider the influence of the feed gap ($g$) between the background and radiation patch, $g$ is incorporated into the height ($h$) of the cylinder, $w$ needs to be revised to $w + g$ in the Formula (2), the lowest resonant frequency ($f_L$) can be approximated by calculating the following [21]:

$$f_L = \frac{144}{w + w' + g + \frac{l}{2\pi\sqrt{\varepsilon_{eff}}} + \frac{l'}{2\pi\sqrt{\varepsilon_{eff}}}}(GHz) \tag{3}$$

At the beginning of the design, Formula (3) can be used to obtain the initial dimensions of the antenna structure, and then simulation software is used to optimize and improve the parameters to improve the design efficiency. The dimensional parameters of Ref. Antenna are obtained as follows: $l = w = 21.8$ mm, $l' = 25.6$ mm, $w' = 4.8$ mm, $g = 1.4$ mm, as shown in Figure 3a. The lowest resonant frequency estimated by Formula (3): $f_L = 3.85$ GHz. It can be seen that the red curve from the reflection coefficient curve given in Figure 4 is indeed resonant around 3.85 GHz, in agreement with the calculation. The purple line in Figure 4 represents $S_{11} = -10$ dB. The operating bandwidth of the reference antenna is only 1.4 GHz (3.07–4.47 GHz), which cannot meet the design requirements of UWB technology, so further improvement is needed.

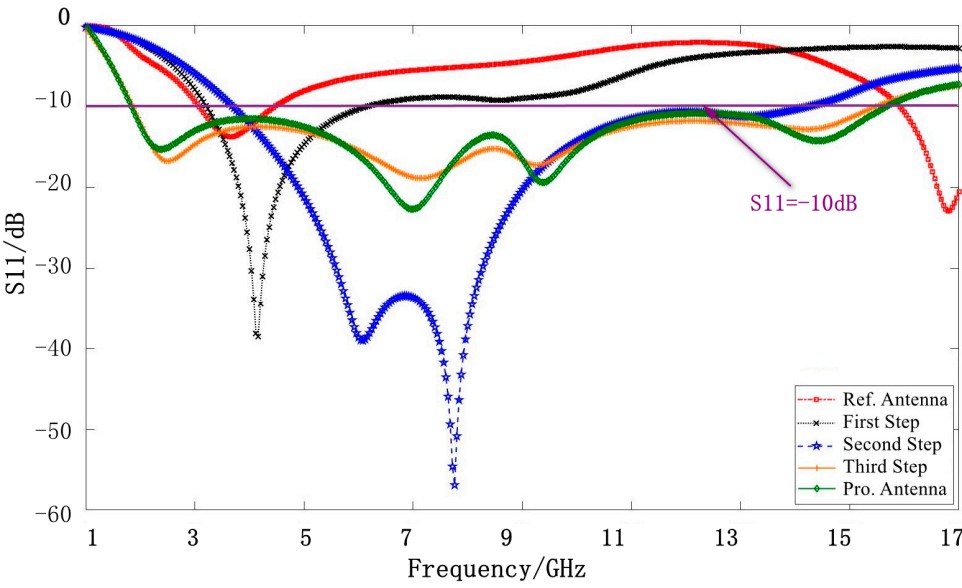

**Figure 4.** Variation curve of the reflection coefficient during the design process.

The coplanar waveguide feeding structure has many advantages over other feeding methods. First, the single-sided structure makes it easy to connect in series or parallel with other microwave devices, facilitating integration with both passive and active microwave devices. Second, the symmetric ground state of CPW also allows for miniaturized circuits and improved radiation patterns. Third, the primary mode transmitted by the coplanar waveguide is a quasi-TEM mode with no minimum resonance frequency, which can transmit both odd and even modes, thus providing new ideas for the design of new antennas. Finally, the dispersion properties of CPW are also higher than those of the microstrip, which is suitable for broadband implementations of microwave circuits and antennas, and for the study of UWB antennas.

Based on the above analysis, it can be seen that the coplanar waveguide feed has many advantages over the microstrip line feed, so the first step of antenna structure improvement is to convert the microstrip line feed into CPW feed, as shown in Figure 3b. The size and position of the feed line and the two grounds in the CPW structure are optimized

and analyzed to achieve a good 50 Ω impedance matching. At this time, the three main parameters affecting the minimum resonant frequency of the antenna (patch length, patch width, patch and ground clearance) are not changed, so the minimum resonant frequency in the working band only has a slight frequency deviation, but the bandwidth has been expanded to a certain extent, the absolute bandwidth is 2.97 GHz (3.21–6.18 GHz), and the relative bandwidth is 63.26%.

The second step is to change the signal line to a stepped feed structure, as shown in Figure 3c, the reduction in the feed gap increases the minimum resonant frequency of the antenna from 3.21 GHz to 3.69 GHz, and also extends the surface current path of the radiating patch, so that the current can flow smoothly in a larger area, which improves the bandwidth characteristics and increases the upper resonant frequency to 14.27 GHz, the absolute bandwidth is 10.58 GHz, the impedance bandwidth reaches 117.82%, and there are two resonant frequencies: 6.08 GHz and 7.74 GHz.

In order to further expand the bandwidth of the antenna, the third step will be two pieces of ground extended to connect together, forming the effect of the surrounding ground, as shown in Figure 3d. At this time the antenna can be approximated as a rectangular slit antenna. The reflection coefficient curve in Figure 4 shows that the antenna works in the range of 1.86–15.49 GHz, which covers the UWB communication band specified by FCC, and there are four resonance points: 2.5 GHz, 7.16 GHz, 9.31 GHz, and 14.32 GHz. The absolute bandwidth of 13.63 GHz and the relative bandwidth of 157.12% is achieved, which is 1.3 times the step CPW-fed antenna (Second Step), 4.6 times the conventional CPW-fed antenna (First Step) and 9.7 times the conventional microstrip wire-fed antenna (Ref. Antenna).

The next step is to embed the "T" shaped metamaterial [22] into the radiating patch, as shown in Figure 3e, which is to optimize the impedance matching and increase the upper resonant frequency of the operating band. In Figure 4, it can be seen that the upper resonant frequency of the operating band is effectively widened to 15.69 GHz, which is 1.8 GHz higher than that in the third step, and there are four resonance points: 2.39 GHz, 7 GHz, 9.4 GHz, and 14.48 GHz, which have a slight frequency shift compared with the four resonance points of the antenna in Figure 3d. The overall operating area of the antenna is 1.83–15.69 GHz, the operating bandwidth reaches 13.86 GHz, and the relative bandwidth is as high as 158.22%. Considering that the antenna sample needs to be connected to the detachable SMA adapter during the laboratory test, the antenna structure in Figure 3e is selected to be perforated at appropriate positions, which will have almost no effect on the performance parameters of the antenna, and finally, the final structure of the UWB monopole antenna is obtained as shown in Figure 3f.

*2.3. Key Parameters Analysis*

In the design process of the UWB antenna, it can be seen from Formula (3) that the main parameters affecting the lowest resonant frequency of the antenna working band are w, fw1, fw2, gw, combined with the simulation software for detailed parameter optimization to get the optimal size, focusing on the influence of w, fw1, fw2, gw on the reflection coefficient, as shown in Figure 5. The green line in Figure 5 represents $S_{11} = -10$ dB.

Since the length and width of the radiating patch of the UWB antenna are close to each other and the structure is approximately square, only one parameter w is discussed. The feed gap depends on the size of the interval between the front patch structure and the reverse ground, which can be calculated by gap = fw1 + fw2 − gw. The antenna size is optimized and fine-tuned by simulation software to obtain the final determined width of the radiating patch w = 10.1 mm, the height of the backside ground gw = 4.81 mm, and the length of the stepped ground fw1 + fw2 = 6.4 mm.

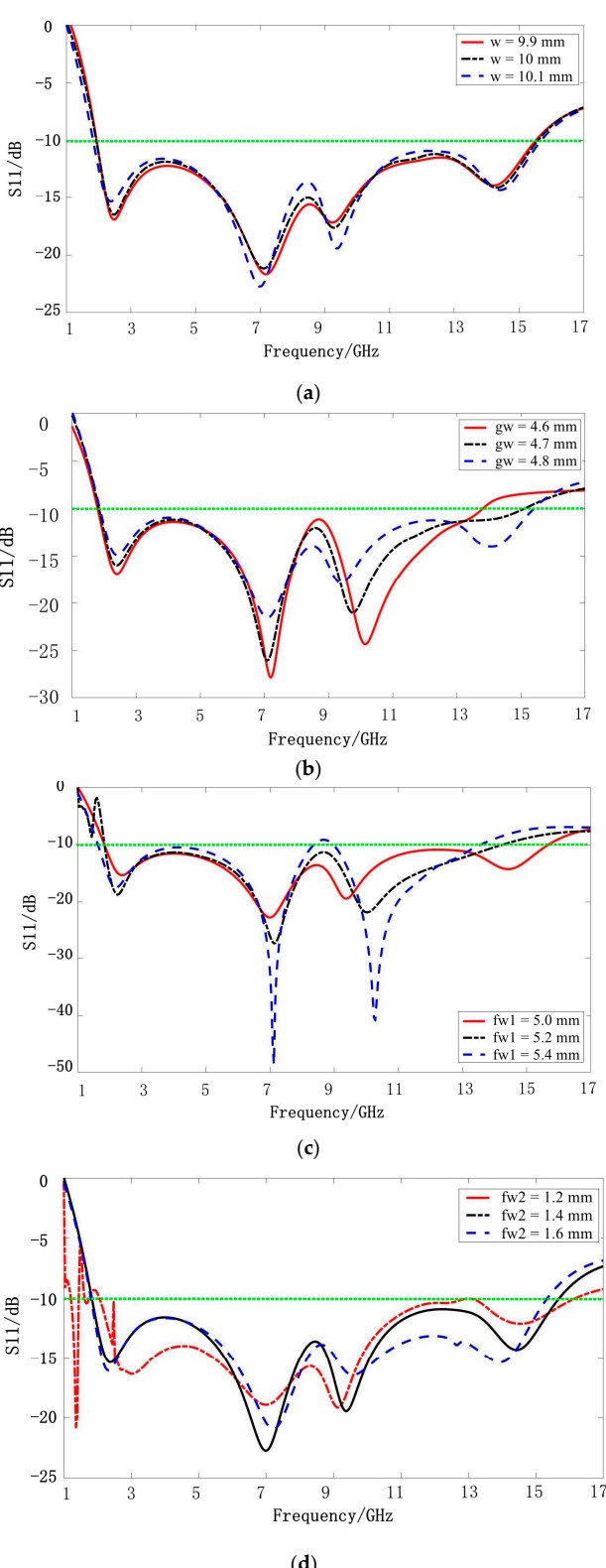

**Figure 5.** The variation curve of the reflection coefficient of UWB antenna with (**a**) w, (**b**) fw1, (**c**) fw2, (**d**) gw.

### 2.4. Simulation Results Analysis

Figure 6 shows the comparison curves between the simulation and test results for the reflection coefficient of the UWB antenna. The green line in Figure 6 represents $S_{11} = -10$ dB. The simulated working area of the antenna is the light green area, from 1.83 GHz to

15.69 GHz, which can completely cover the 3.1–10.6 GHz required for UWB communication. The blue dashed line is the $S_{11}$ curve obtained by the vector network analyzer. The measured $S_{11} < -10$ dB band is 1.8–16.38 GHz, the absolute bandwidth reaches 14.58 GHz and the relative bandwidth is up to 160.4%. At the same time, it can be observed that the test curve of the reflection coefficient basically matches the trend of the simulation curve, and the working band has been expanded to a certain extent, which is related to the loss compensation function of the vector network itself, and the error is mainly due to the manual welding of the antenna sample adapter, the slight movement of the antenna sample during the test, and the wear and tear of the test connection itself.

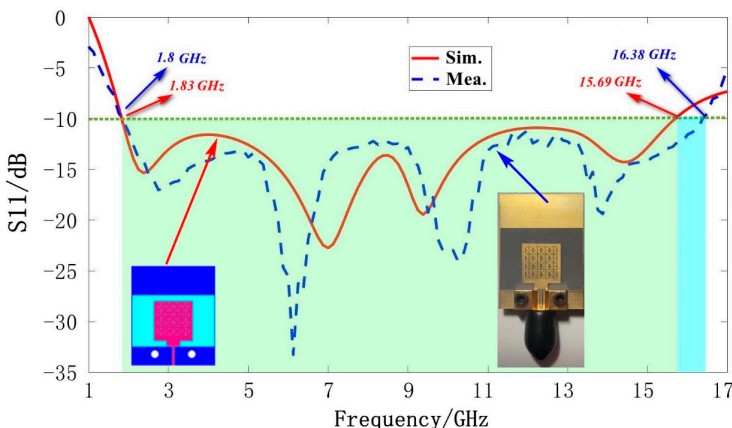

**Figure 6.** Comparison of simulation and test curves for $S_{11}$ of the UWB antenna.

The radiation characteristics of the UWB antenna are viewed by the simulation software HFSS, and the radiation patterns of the E-plane (XOZ plane) and H-plane (XOY plane) at four frequency points: 3 GHz, 6.16 GHz, 10.6 GHz and 14.92 GHz are shown in Figure 7. As shown in Figure 7a, the radiation patterns in the E-plane of this UWB antenna are all approximately circular, showing stable omnidirectional radiation characteristics. The radiation pattern in the H-plane shown in Figure 7b approximately shows certain bidirectional radiation characteristics, but with the increase in frequency, there is a certain distortion, especially at the frequency of 14.92 GHz, which is due to the fact that with the increase in frequency, the wavelength gradually becomes smaller and the antenna size can be compared with it, and the radiation pattern is more easily affected.

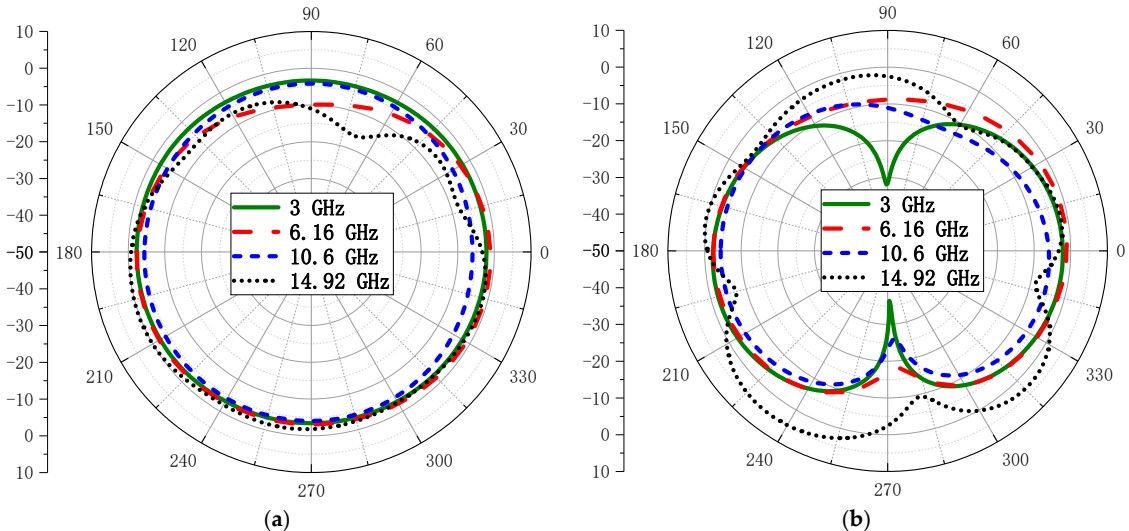

**Figure 7.** Simulated Radiation pattern of UWB monopole antenna in (**a**) E-plane ($\varphi = 0^{\circ}$) (**b**) H-plane ($\varphi = 90^{\circ}$).

## 3. Quad-Port UWB MIMO Antenna Design

To meet the channel capacity requirements in modern wireless communications, a compact low-coupling four-cell UWB MIMO antenna is designed in this section. Only four UWB antennas are required as fundamental units, and the effectiveness of the decoupling is achieved by using polarization diversity and introducing a decoupling gap in the metallic ground while keeping the overall dimensionality unchanged. Isolation can be reflected in transmission and correlation coefficients. To further validate the effectiveness of this decoupling combination, we performed a sample and test. The test curves are in general in good agreement with the simulations, and the slight deviations are mainly due to manual manipulation, within the error tolerance.

### 3.1. MIMO Antenna Structure

The UWB antenna is used as a unit cell and the four cells are rotated symmetrically at the center by the polarization diversity technique, introducing four decoupled slits on the metallic ground to form the UWB MIMO antenna in Figure 8. The quad-port MIMO antenna is printed on the single side of the F4B substrate and the overall size is 51.2 mm × 51.2 mm × 1.524 mm.

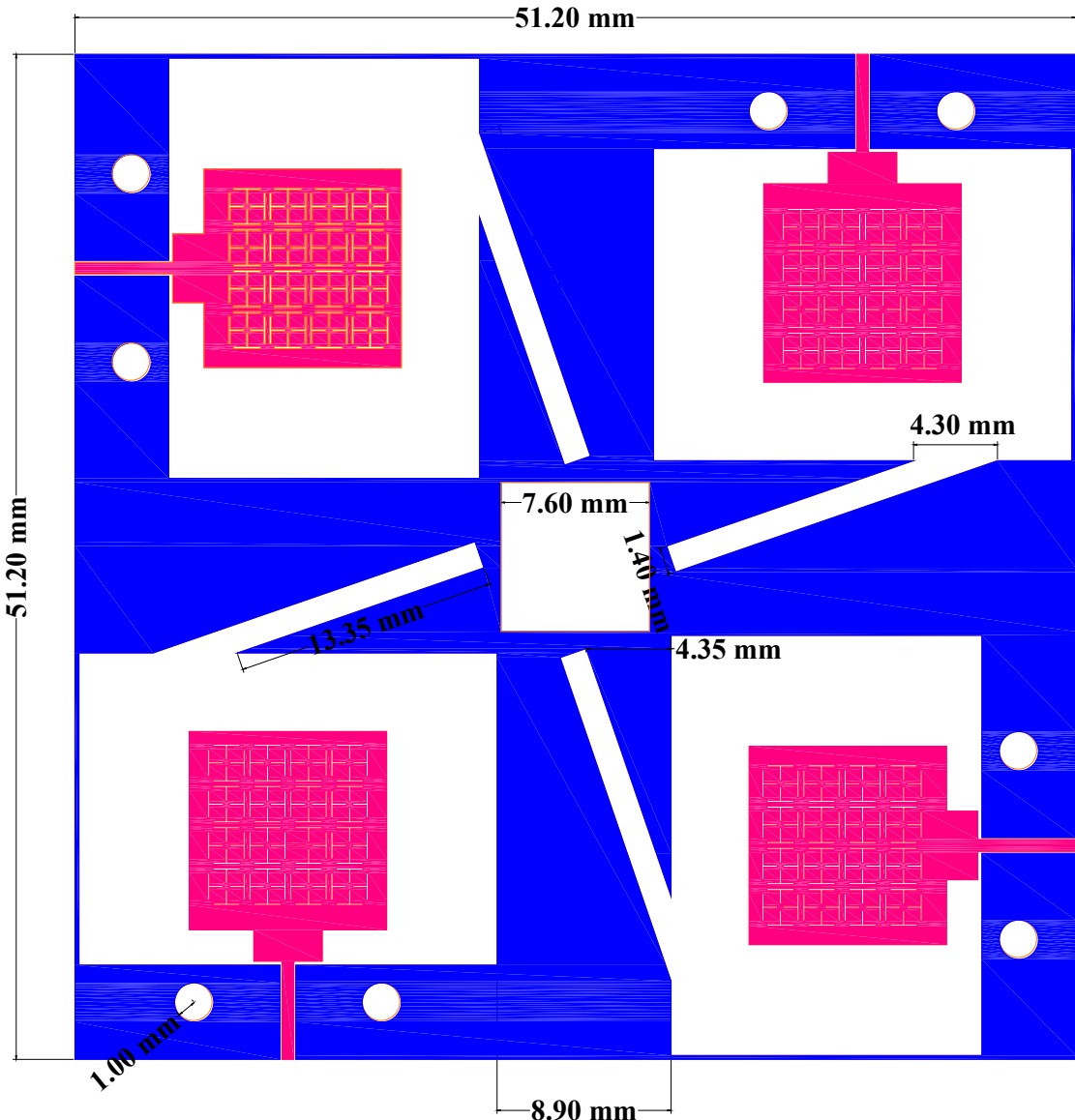

**Figure 8.** Structure of a four-cell MIMO antenna with polarization diversity.

The layout of MIMO antennas with polarization diversity makes it easier to achieve high isolation, which is due to the fact that the antenna elements with linear polarization are perpendicular to each other, and the coupling is lower since the polarization methods between the elements are independent of each other. Without an additional decoupling device, a polarization diversity technique and a slit in the metallic ground state are used to reduce the coupling The effectiveness of this method is mainly due to the fact that the introduction of the slit destroys the integrity of the metal ground, changes the current transmission path, and to some extent truncates the coupling path of part of the current between the units, so the isolation degree can be improved.

### 3.2. S-Parameters Analysis

Without sacrificing the size of the MIMO antenna, the isolation of the antenna is substantially improved, and the effectiveness of the decoupling can be verified by the transmission coefficient between the four ports of the antenna. One unit of the four-cell UWB MIMO antenna in Figure 8 is excited individually, and the remaining ports are matched with 50 Ω load to check the reflection coefficient and transmission coefficient of the four ports, respectively.

From Figure 9a,b, it can be seen that there is a small shift in the reflection coefficient for different end ports of the antenna at lower and higher frequencies, while the other regions can be well matched. Four ports are in the same position, so Figure 9a shows the port reflection coefficient: $S_{11} \approx S_{22} \approx S_{33} \approx S_{44}$, Figure 9b shows the transmission coefficient between adjacent ports: $S_{12} \approx S_{23} \approx S_{34} \approx S_{41}$, the transmission coefficient between two ports spaced by 2: $S_{13} \approx S_{24}$, these can be observed in Figure 9.

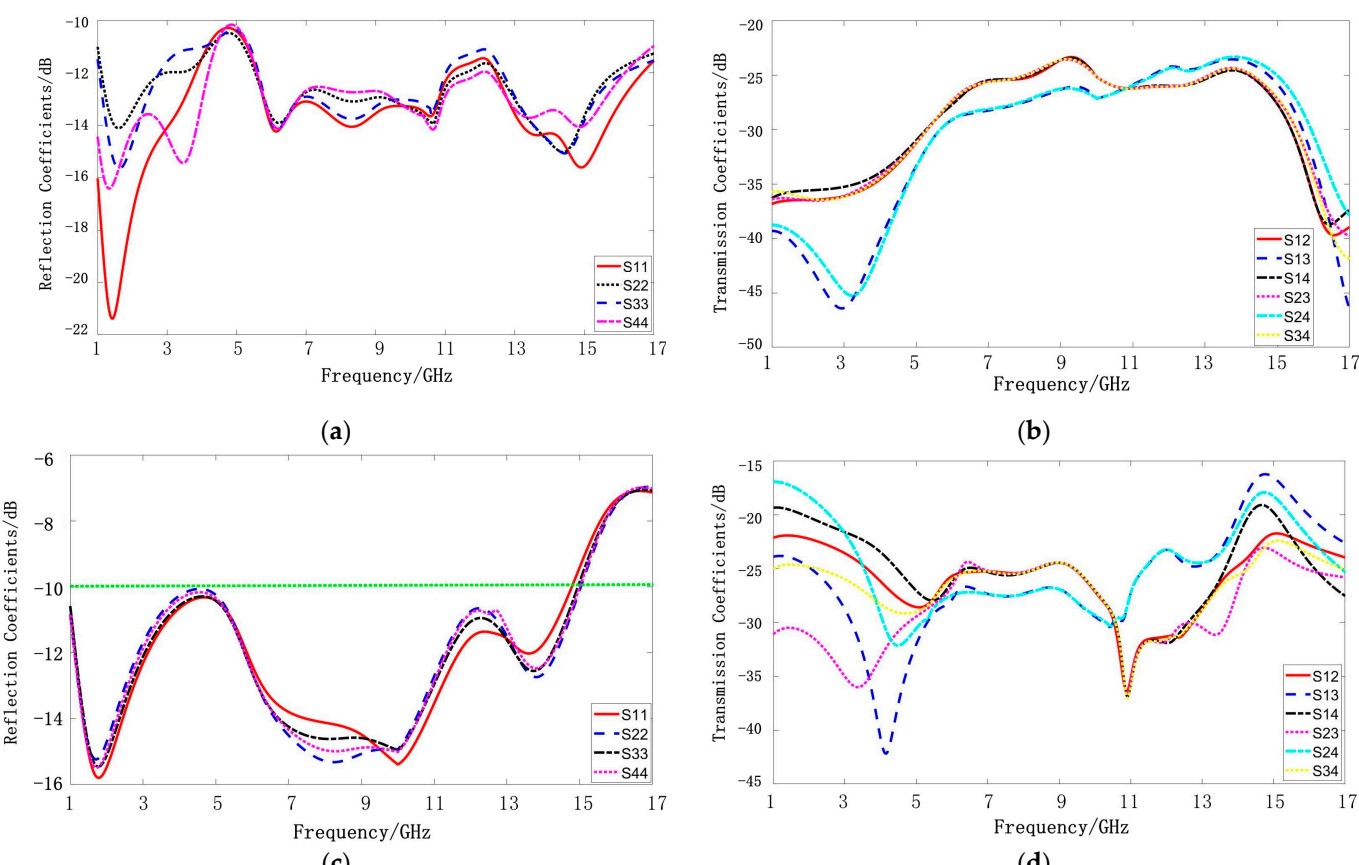

**Figure 9.** S-parameters of the four ports of the UWB MIMO antenna: (**a**) reflection coefficient with strips; (**b**) transmission coefficient with strips; (**c**) reflection coefficient without strips; (**d**) transmission coefficient without strips.

To illustrate the role of introducing strips in reducing the coupling, the S-parameter simulation results without introducing strips are given in Figure 9c,d. The green line in Figure 9 represents S$_{11}$ = −10 dB. Comparing Figure 9a,c, it can be seen that the reflection coefficients at each port of the MIMO antenna have better similarity when the strips are not introduced, and the return loss depth of the antenna at low frequencies is deepened and the impedance bandwidth is expanded to some extent after introducing strips. Comparing Figure 9b,d, it can be seen that the isolation of the antenna at both lower and higher frequencies is improved after introducing strips, especially at <4 GHz and >14 GHz. These are both observed in Figure 9, indicating that the coupling can indeed be reduced by introducing strips.

## 4. Result and Discussion

### 4.1. S-Parameters

Due to the quasi-similarity of the four cells of the UWB MIMO antenna, the reflection and transmission coefficients with only one port were chosen for the study, and the simulation and test results were analyzed and compared as shown in Figure 10.

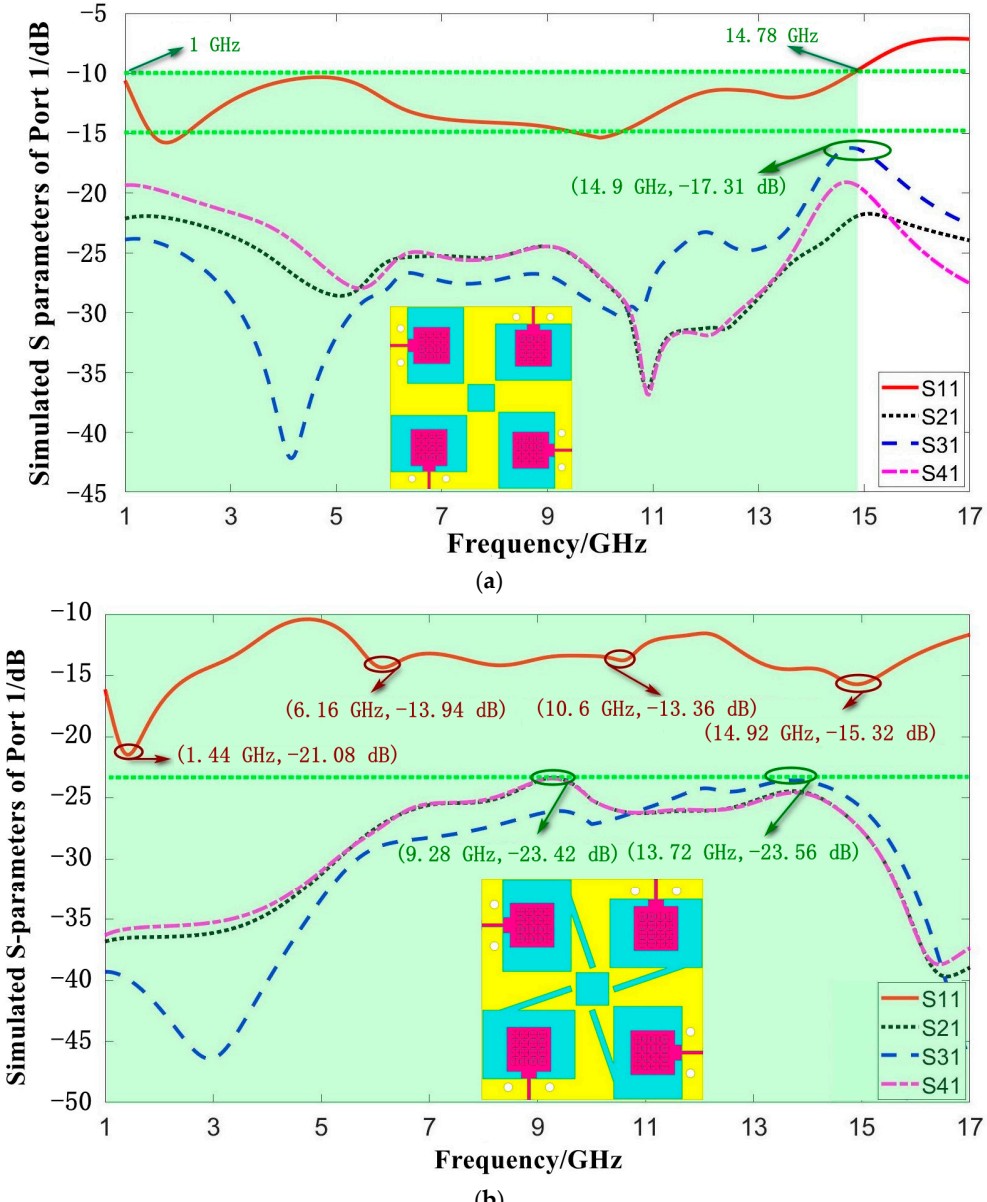

(a)

(b)

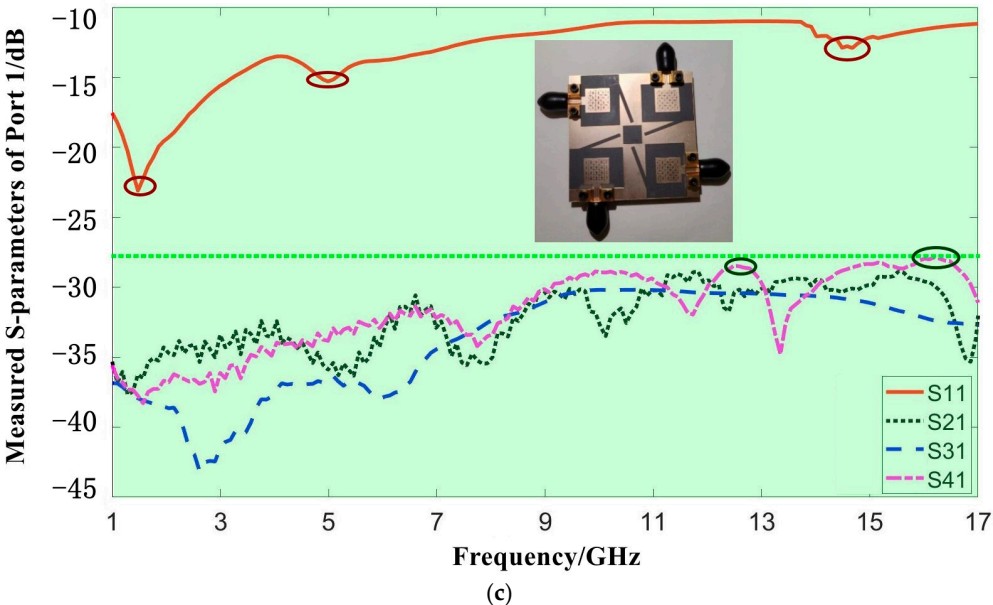

(c)

**Figure 10.** Reflection coefficient and transmission coefficient of UWB MIMO antenna at port 1: (**a**) simulation without introduced strips; (**b**) simulated results with strips; (**c**) measured results with strips.

In order to analyze and illustrate the decoupling effect of the MIMO antenna with the introduction of the strips, the S-parameters of the MIMO antenna at port 1 before the inclusion of the gap are also given, as shown in Figure 10a. The two green dashed lines in Figure 10a represent $-10$ dB and $-15$ dB, respectively, and are marked in the figure for a more visual observation of the operating area of the antenna ($S_{11} \leq -10$ dB, $S_{21}$, $S_{31}$, $S_{41} \leq -15$ dB). The green area is the $-10$ dB impedance bandwidth of the antenna: 1–14.78 GHz, which can cover the UWB communication range of 3.1–10.6 GHz with an absolute bandwidth of 13.78 GHz and a relative bandwidth of 174.65%. It can be seen that the isolations degree between port 1 and the other three ports is $\geq 17.31$ dB, which meets the requirements of the MIMO antenna for isolation degree. When the antenna is operating at 14.9 GHz, the isolation between ports 1 and 3 has the lowest value of 17.31 dB, which indicates that the diversity of polarization modes can effectively improve the coupling of MIMO antennas, but the improvement results have some limitations.

The simulated S-parameters of the MIMO antenna at port 1 after adding the decoupling slit are shown in Figure 10b, as can be seen from the reflection coefficient curve, the antenna operates at 1–17 GHz with an absolute bandwidth of 16 GHz and a relative bandwidth of 177.78%. The green dashed line in Figure 10b,c represent the maximum value of the transmission coefficients in the working area, which can be judged as the minimum value of the isolation. From the transmission coefficient, it can be seen that the isolation between the ports all exceeds 23.42 dB, and the isolation effect is improved, especially when the antenna works below 4 GHz and above 14 GHz, the isolation degree is improved obviously, and the simulated isolation degree between different ports is shown in Table 2.

**Table 2.** The simulated isolation of the proposed UWB MIMO antenna.

| Isolation | Minimum/dB | Maximum/dB |
|---|---|---|
| Port 1 to 2 | 23.72 | 39.27 |
| Port 1 to 3 | 23.46 | 46.77 |
| Port 1 to 4 | 23.71 | 37.14 |

Figure 10c shows the measurement results after the MIMO antenna is introduced strips. The measured results show that the curve trends of the S-parameter measured

and simulated are basically matched, with a small offset due to manual operation within the error range. The test data show that the MIMO antenna meets: $S_{11} < -10$ dB and $S_{21}, S_{31}, S_{41} \leq -27.8$ dB in the operation band of 1–17 GHz. The measured S-parameter of the proposed antenna also indicates that this antenna is a UWB MIMO antenna with high isolation and good performance. The proposed MIMO antenna can be used not only in ultra-broadband communication (3.1–10.6 GHz) systems, but also can be used for satellite mobile communication in a lower frequency band (1–3.1 GHz) and for fixed communication in the higher frequency band (10.6–17 GHz).

### 4.2. Far-Field Characteristics

As shown in Figures 11 and 12, the radiation pattern in E-plane (XOZ), H-plane (XOY), and 3D radiation pattern at four frequency points (3 GHz, 6.16 GHz, 10.6 GHz, 14.92 GHz) are selected to study the far-field radiation characteristics of proposed MIMO antenna.

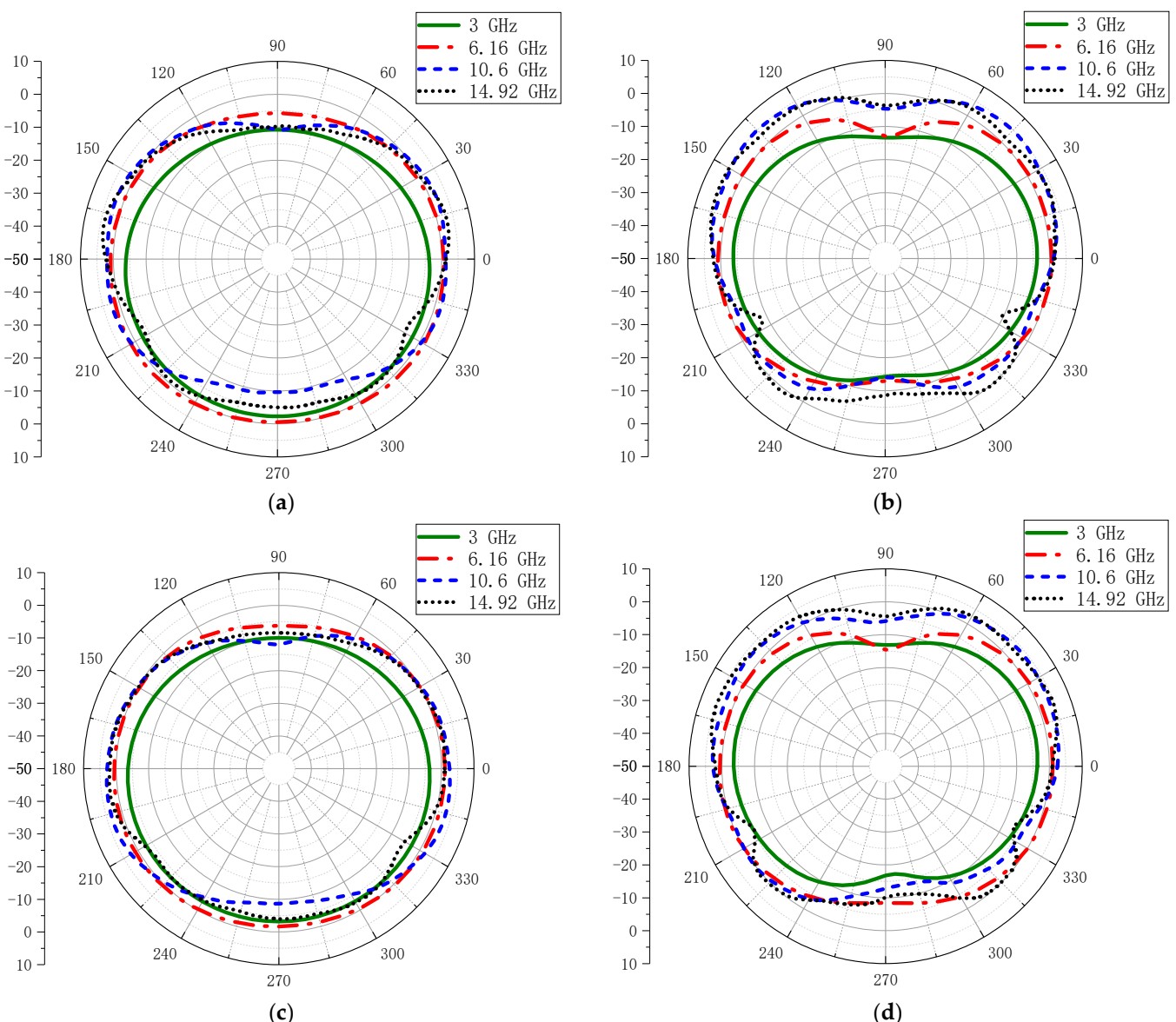

**Figure 11.** Simulated radiation pattern of proposed MIMO antenna in: (**a**) E-plane (without strips); (**b**) H-plane (without strips); (**c**) E-plane (with strips); (**d**) H-plane (with strips).

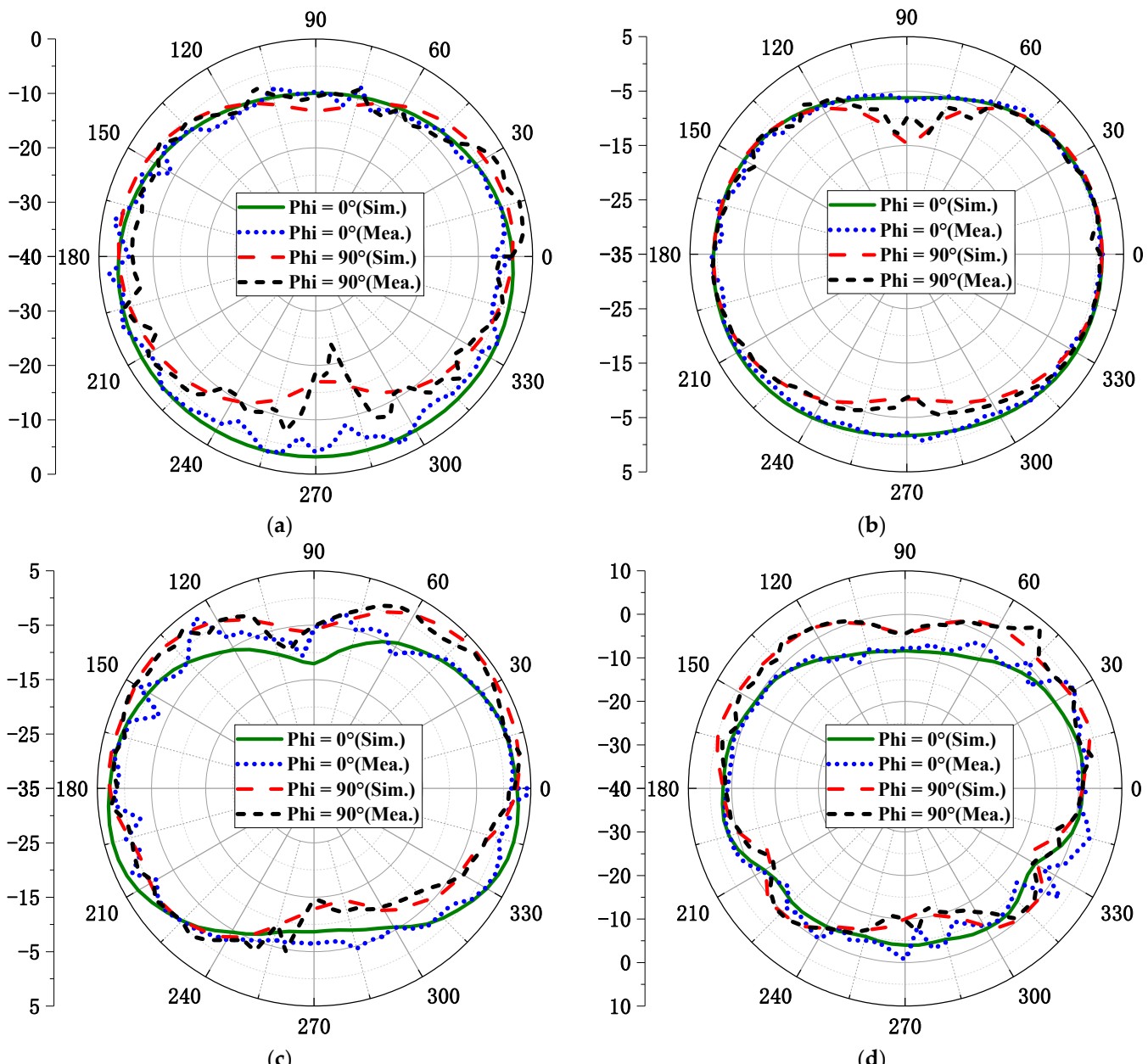

**Figure 12.** Simulated and measured radiation patterns of the proposed MIMO antenna in two main planes at: (**a**) 3 GHz; (**b**) 6.16 GHz; (**c**) 10.6 GHz; (**d**) 14.92 GHz.

The first consideration is whether the introduction of the strip to reduce the coupling will affect the far-field characteristics of the antenna, so the simulation radiation patterns of the antenna in two main planes at the four frequency points without and with introducing strips are viewed, respectively, as shown in Figure 11. The radiation patterns in E-plane and H-plane are shown in Figure 11a,b, when the metal strips are not introduced, and in Figure 11c,d when strips are introduced. It can be observed that the introduction of the strip has almost no effect on the radiation properties of the proposed MIMO antenna.

The radiation pattern of the proposed MIMO antenna at the four frequency points is shown in Figure 12, including the measured and simulated results in the E and H planes. From Figure 12a, it can be seen that when the antenna works at 3 GHz, the antenna radiates omnidirectionally in the E plane, and shows slight directional radiation characteristics along 0° and 180° in the H plane. Figure 12b shows that the MIMO antenna shows similar radiation characteristics at 6.16 GHz of 3 GHz. The radiation patterns in Figure 12c,d are

slightly distorted, but also have basically omnidirectional radiation characteristics. At 10.6 GHz, the radiation pattern in the E-plane is collapsed at 90° in Figure 12c. When the antenna operates at 14.92 GHz, the E-plane in Figure 12d collapses near 210° and 330°, and the H-plane also decreases sharply near 94° and 288°, mainly because the parameters at high frequencies are more sensitive to size and more susceptible to influence.

By comparing the measurement results and simulation results of the radiation patterns of the proposed MIMO antenna in two main planes in Figure 12, it can be seen that the overall trend of both is basically the same, and the reason for many burrs in the measured results may be due to manual testing and mechanical jitter. Comparing and analyzing the measurement and simulation results, the radiation pattern of the proposed MIMO antenna is similar to a circle in the E plane, which can radiate omnidirectionally, and has a slight directional capability in the H plane, which is more obvious at low frequencies and gradually weakened at high frequencies.

Figure 13 gives a 3D radiation pattern for these four frequency points, which allows one to visualize the radiation properties of the antenna. Figure 14 shows the gain and radiation efficiency of the MIMO antenna for the two cases with and without the addition of the metal strip. It can be seen that the values of the gain and efficiency of the MIMO antennas are almost unaffected by the introduction of the strip to increase the isolation, and the trends of the variations almost coincide. The gain and radiation efficiency of the proposed MIMO antennas are favorable for the whole operating band. The gain steadily increases to a maximum of 7.81 dB and the radiative efficiency is always stable between 90% and 98%.

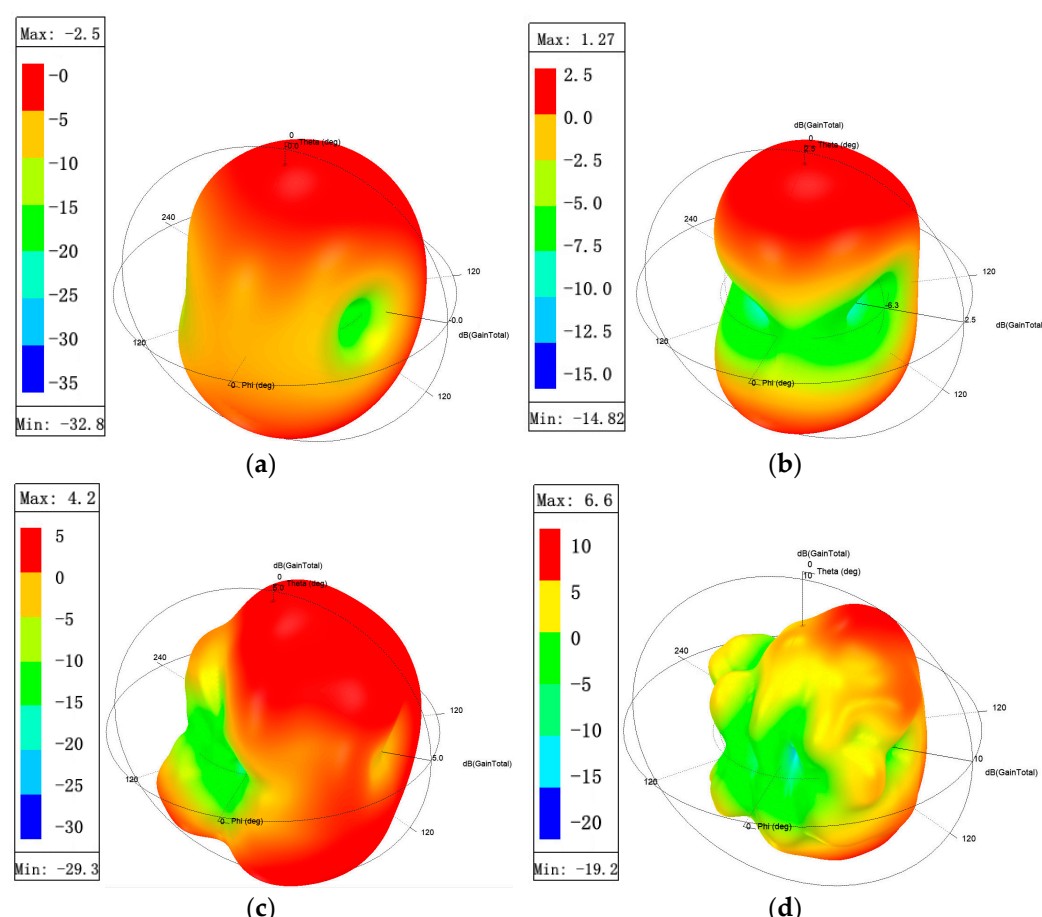

**Figure 13.** Three-dimensional radiation patterns of proposed MIMO antenna at: (**a**) 3 GHz; (**b**) 6.16 GHz; (**c**) 10.6 GHz; (**d**) 14.92 GHz.

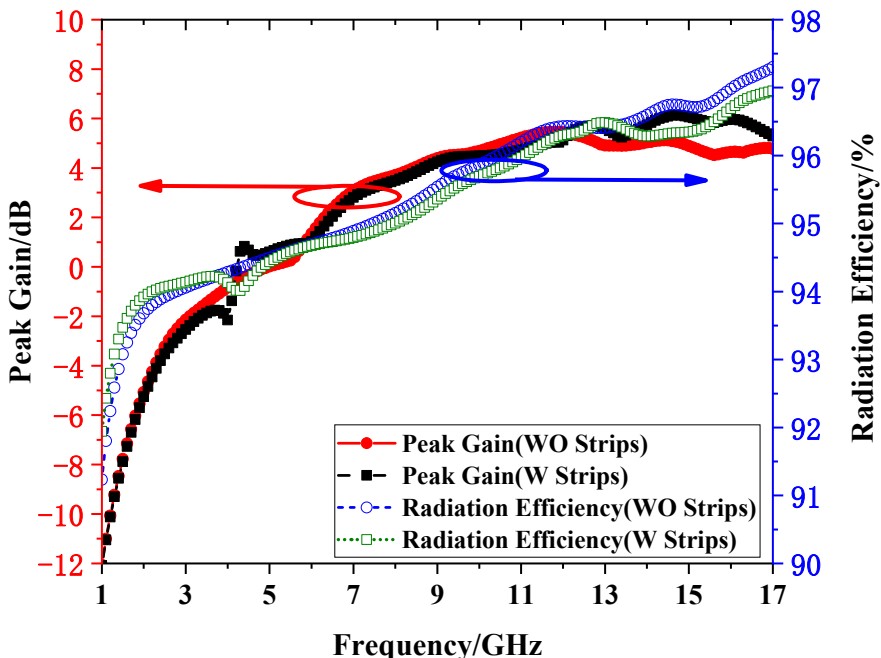

**Figure 14.** Gain and radiation efficiency of UWB MIMO antenna with/without introduced strips.

### 4.3. Diversity Properties of MIMO Antennas

In this section, the diversity characteristics of the proposed UWB MIMO antenna are analyzed by the Envelope Correlation Coefficient (ECC), Diversity Gain (DG), Mean Effective Gain (MEG) and Total Active Reflection Coefficient (TRAC).

#### 4.3.1. Envelope Correlation Coefficient

To see the diversity properties of MIMO antennas, the envelope correlation coefficient ECC is calculated using the S-parameter. The ECC curves between ports 1 and 2 and the other ports in the MIMO antenna are shown in Figure 15a. Figure 15b shows the ECC curves between the remaining three ports as a function of frequency.

$$ECC = \frac{\left| S_{ii}^* \times S_{ij} + S_{ji}^* \times S_{jj} \right|^2}{\left(1 - S_{ii}^2 - S_{ij}^2\right)\left(1 - S_{ij}^2 - S_{jj}^2\right)} \tag{4}$$

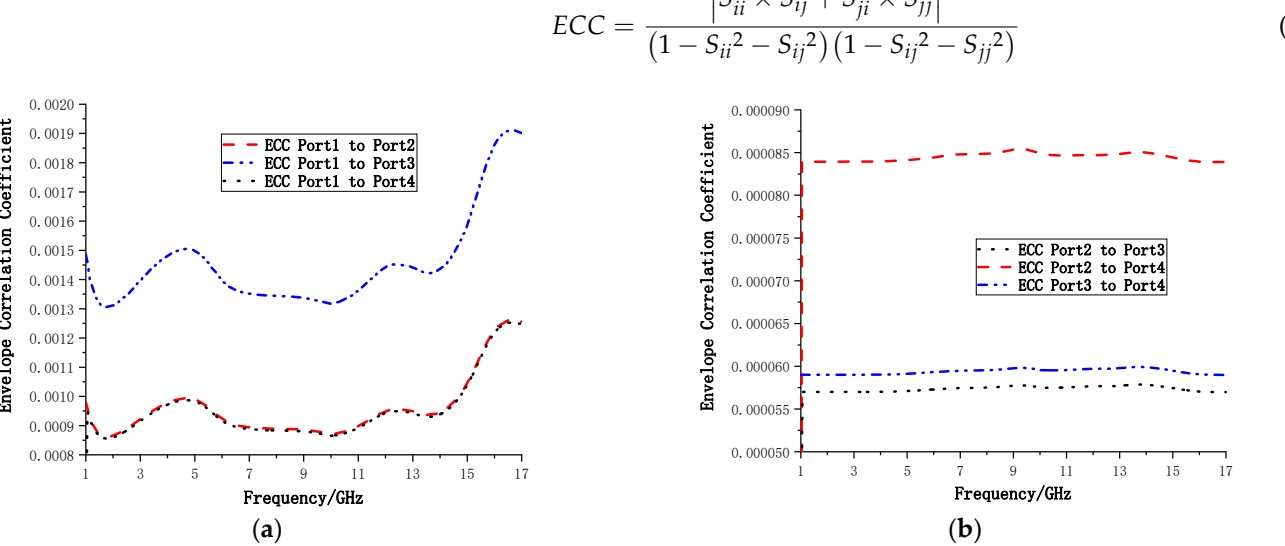

**Figure 15.** ECC between antenna units: (**a**) between port 1 and other ports; (**b**) between ports 2, 3, and 4.

From Figure 15a, it can be seen that the ECC of port 1 to port 2 (ECC12) and port 1 to port 3 (ECC13), and port 1 to port 4 (ECC14) is less than 0.002 within the operating frequency, especially the ECC between ports 1, 3, and 1,4 are less than 0.0013. In Figure 15b, it can be seen that the ECC of ports 2, 3, and 4 is less than 0.00009, and the ECC of port 2 to port 3 (ECC23) and port 2 to 4 (ECC24) is less than 0.00006, and the ECC of all ports is less than 0.002, which satisfies the requirements of MIMO system for MIMO antenna correlation parameters [23].

### 4.3.2. Diversity Gain

Another important parameter describing the diversity characteristics of the MIMO system is the Diversity Gain (DG), which can be calculated by ECC [24].

$$DG = 10 \times \sqrt{1 - |ECC|^2} \tag{5}$$

Figure 16 shows the diversity gain of the proposed antenna at ports 1 and 2 is greater than 9.99 dB, which indicates that the MIMO antennas have good diversity properties.

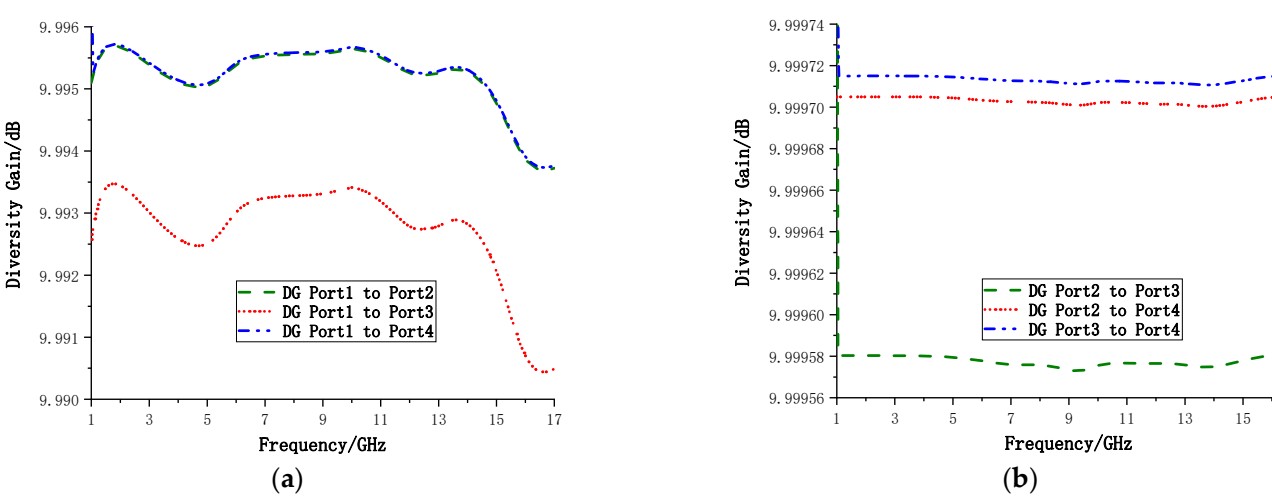

**Figure 16.** Diversity Gain curve of proposed antenna at: (**a**) port 1; (**b**) port 2.

### 4.3.3. Mean Effective Gain

In a multipath fading environment, the Mean Effective Gain (MEG) can be used to measure the total efficiency of the MIMO system, the total gain, and the effect produced by the propagation environment [25]. It is generally required that the MEG between the units within the MIMO system is less than 3 dB, and the MEG can be estimated from the S-parameters of each port [26].

$$MEG_i = 0.5 \times \left(1 - \sum_{j=1}^{4} S_{ij}\right)(i = 1, 2, 3, 4) \tag{6}$$

$$MEG_{ij} = MEG_i - MEG_j (i \neq j) \tag{7}$$

The $MEG_1$, $MEG_2$, $MEG_3$ and $MEG_4$ at the four ports calculated by formula (6) are all less than 6 dB, as shown in Figure 17a–c. Similarly, $MEG_{12}$, $MEG_{13}$ and $MEG_{14}$ calculated by Formula (7) are all between −0.2 dB and 0.2 dB, and the data details are shown in Figure 17d; it can be seen that the average effective gain of the proposed MIMO antenna performs well over the whole operating band.

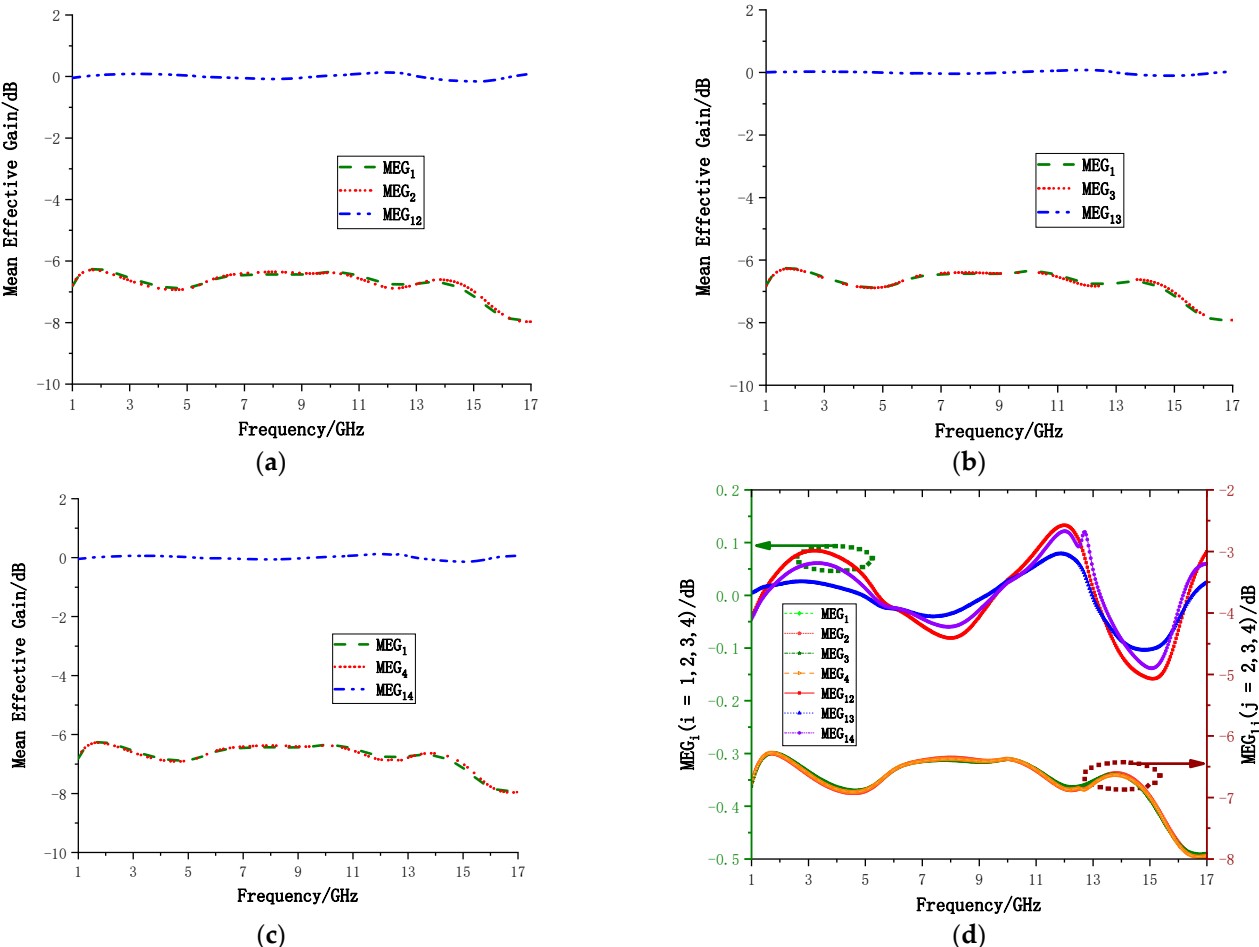

**Figure 17.** Mean Effective Gain of proposed MIMO antenna between: (**a**) port 1 and 2; (**b**) port 1 and 3; (**c**) port 1 and 4; (**d**) details.

### 4.3.4. Total Active Reflection Coefficient

In order to describe more rigorously the effect of influence between adjacent cells in MIMO systems, the Total Active Reflection Coefficient (TARC) is used to describe the coupling degree and channel independence between ports, in addition to the S-parameters of individual ports [27,28].

$$TARC = -\sqrt{\frac{(S_{11} + S_{1i})^2 + (S_{i1} + S_{ii})^2}{2}}, \ (i = 2, 3, 4) \tag{8}$$

In the ultra-wideband range, the TARC is guaranteed to be $<-10$ dB at a minimum [29]. It can be seen from Figure 18 that the TARC from port 1 to port 2 is between $-52.28$ dB and $-36.34$ dB. The TARC from port 1 to port 3 is between $-59.13$ dB and $-35.08$ dB, and the TARC from port 1 to port 4 is between $-51.46$ dB and $-36.41$ dB. All TARCs are below $-35$ dB, which ensures low coupling effects and channel independence between receiver and transmitter in MIMO systems.

To further illustrate the advantages of the proposed MIMO antennas, recent designs of MIMO antennas are compared with UWB MIMO antennas. All antennas are listed in Table 3 in terms of operational bandwidth, isolation, correlation coefficient, and size. The port isolation of antennas in [3–5,7,15,16] is more than 24 dB, but bandwidth is limited and only two cells, while the proposed antenna has a wider bandwidth with four cells and port isolation > 27.8 dB. The antennas in [10,11,19] all have four cells and can completely include the UWB band (3.1–10.8 GHz), but the size of the antennas is generally large, which

is not conducive to the miniaturization of MIMO antennas. The isolation of [11,12,15] is low, in particular, [11] also has the problem of being oversized. The comparison in Table 3 shows that the proposed four-cell MIMO antenna has the advantages of wide bandwidth, high isolation, low correlation, miniaturization and compact structure.

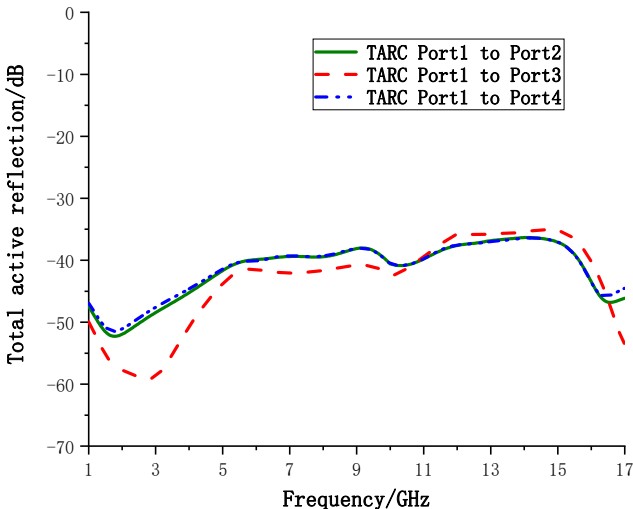

**Figure 18.** The TARC curves for the UWB MIMO antenna.

**Table 3.** Compares the proposed antenna with other MIMO antennas in the references.

| Ref. | Operation Band/GHz | Isolation/dB | ECC | Size/mm$^2$ | Unit No. | Unit Size/mm$^2$ |
|------|--------------------|--------------|-----|-------------|----------|------------------|
| [3] | 2.33–6.71 | >24 | <0.036 | 56 × 70 | 2 | 1960 |
| [4] | 3–11 | >25 | <0.004 | 25 × 35 | 2 | 875 |
| [5] | 3.1–10.6 | >31 | <0.035 | 47 × 93 | 2 | 2185.5 |
| [7] | 3–10.7 | >25 | ≤0.01 | 34.9 × 43 | 2 | 750.35 |
| [10] | 2.1–20 | >25 | <0.02 | 80 × 80 | 4 | 1600 |
| [11] | 2.8–13.3 | >18 | <0.06 | 72 × 72 | 4 | 1296 |
| [12] | 3–13.5 | >15 | <0.4 | 40 × 40 | 4 | 400 |
| [15] | 4.3–11.6 | >15.8 | <0.007 | 41 × 99.4 | 2 | 2037.7 |
| [16] | 2.7–11.22 | >20 | <0.035 | 30.75 × 37.8 | 2 | 581.175 |
| [19] | 2.75–16.05 | >20 | <0.006 | 55 × 55 | 4 | 756.25 |
| Pro. | 1–17 | >27.8 | <0.002 | 51.2 × 51.2 | 4 | 655.36 |

## 5. Conclusions

In this paper, we design a compact four-cell MIMO antenna with high isolation and ultra-wideband capability. The MIMO antenna consists of four rectangular patch antenna units in an orthogonal layout using an encircling ground, with four inclined slits introduced on the ground between the units, and the overall size is 51.2 mm × 51.2 mm × 1.524 mm. By manipulating the metamaterial structure on the ground and etched patch, the bandwidth of the MIMO antenna is extended to 1–17 GHz, covering multiple bands (L-band, S-band, C-band, X-band and most of Ku-band) with an absolute bandwidth of 16 GHz and a relative bandwidth of 177.78%. Without sacrificing the overall antenna size, the isolation of the proposed UWB MIMO antennas is increased to more than 27.8 dB by using a combination of polarization diversity techniques and introducing gaps in the patches. The measured results show that the proposed antenna has a stable omnidirectional radiation pattern, good gain and efficiency, and is a high isolation and ultra-wideband four-cell MIMO antenna. Meanwhile, the MIMO antenna has good diversity characteristics with acceptable ECC (<0.002), DG (>9.99 dB), MEG (<−6 dB) and TARC (<−35 dB), indicating that it is a MIMO antenna with excellent performance for UWB communication, and the excellent broadband

capability means the antenna can be applied to satellite mobile communication systems and fixed communication systems as well.

**Author Contributions:** Conceptualization, Z.H. and J.J.; methodology, Z.H.; software, Z.H.; validation, Z.H.; formal analysis, Z.H.; data curation, Z.H.; writing—original draft preparation, Z.H.; writing—review and editing, J.J.; supervision, J.J.; project administration, J.J.; funding acquisition, J.J. All authors have read and agreed to the published version of the manuscript.

**Funding:** This research was funded by the National Natural Science Foundation of China (no. 61571320, 62075163).

**Data Availability Statement:** Not applicable.

**Conflicts of Interest:** The authors declare no conflict of interest.

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
