# Peer review of "Compact Quad-Port MIMO Antenna with Ultra-Wideband and High Isolation"

_electronics, doi:10.3390/electronics11203408_

Round 1

Reviewer 1 Report

Authors have presented work on compact quad-port MIMO antenna with ultra-wideband and high isolation. The antenna elements have metamaterial structure on the radiating patch with a coplanar waveguide feed and has broadband characteristics. To reduce the coupling between cells, four angled slits are placed in common ground to improve the isolation of the MIMO antennas. MIMO performance parameters have been investigated. Following comments will be helpful in improving the manuscript.

Line 191: check the variable, some might be missing.

Line 218: replace “manual welding” with “manual soldering”.

Leave line gap between captions of figures and following paragraphs.

Radiation characteristics discussing is using planes i.e. YoZ, XoZ plane etc, however the axis information is missing in design figures, please add axis information to design figures.

Figure 7 and 11 shows radiation characteristics at 1.44 GHz, however the operating band starts from 1.8 GHz. Please justify or replace 1.44GHz with 3 GHz.

Clearly mention in caption of Fig. 7 if the results are simulated or measured. Like its mentioned in Fig. 11.

In caption of Fig 11, replace “tested” with “measured”.

Colour scale values text is not readable. Please use a larger font size and clear pictures. Also replace 1.44 GHz pattern with 3 GHz.

Overall work is good and is well presented.

Author Response

Dear Reviewer 1,

      Thank you very much for your time involved in reviewing the manuscript “Compact Quad-port MIMO Antenna with Ultra-wideband and High Isolation” (ID: electronics-1977586). And your very encouraging comments on the merits. 

      We also appreciate your clear and detailed feedback and hope that the explanation adequately addresses all your concerns. In the remainder of this letter, we discuss each of your comments individually, along with our corresponding responses.

     To facilitate this discussion, we first re-type your comments in italics, and then present our responses to the comments. Careful modifications have been made to our manuscript, all of which are marked in red text, and the screenshot is in the attachment.

Comment 1:

 Line 191: check the variable, some might be missing.

Response 1:

Many thanks for your careful advice. We have changed the variables to "w, fw1, fw2, gw" in line 194 (formerly line 191).

Comment 2:

Line 218: replace “manual welding” with “manual soldering”.

Response 2:

Many thanks for your careful advice. We have changed "manual welding" to "manual soldering" in line 221 (formerly line 218).

Comment 3:

Leave line gap between captions of figures and following paragraphs.

Response 3:

Thank you very much for your careful advice. We have added blank lines before and after each image and table.

Comment 4:

Radiation characteristics discussing is using planes i.e. YoZ, XoZ plane etc, however the axis information is missing in design figures, please add axis information to design figures.

Response 4:

Thanks so much for the reminder. That's a lot of value to us. We have added information about the axes in Figure 1 (a) and made sure that the E plane is the XOZ plane and the H plane is the XOY plane in the full text with changes in rows 227 and 331.

Comment 5:

Figure 7 and 11 shows radiation characteristics at 1.44 GHz, however the operating band starts from 1.8 GHz. Please justify or replace 1.44GHz with 3 GHz.

Response 5:

Thank you so much for pointing this out. This is particularly helpful. We have re-simulated and tested the radiation direction map of the antenna at 3 GHz, replacing 1.44 GHz, and updated the data and redrawn Figs. 7, 11, 12, and 13.

Comment 6:

Clearly mention in caption of Fig. 7 if the results are simulated or measured. Like its mentioned in Fig. 11.

Response 6:

Thanks so much for the reminder. We have changed the title of Fig. 7 to: Simulated Radiation pattern of UWB monopole antenna in (a) E-plane (φ=0°) (b) H-plane (φ=90°).

Comment 7:

In caption of Fig 11, replace “tested” with “measured”.

Response 7:

Thank you very much for your careful suggestion. We have changed "tested" to "measured" in the title of Figure 11.

Comment 8:

Colour scale values text is not readable. Please use a larger font size and clear pictures. Also replace 1.44 GHz pattern with 3 GHz.

Response 8:

Thank you very much for your careful suggestion. We have redrawn the 3D radiation pattern in Figure 13 with increased font size and clarity in the hope that it will meet your reading needs. The frequency point in Figure 13 (a) has been updated from 1.44GHz to 3GHz.

       We would like to take this opportunity to thank you for all your time involved and this great opportunity for us to improve the manuscript. We hope you will find this revised version satisfactory.

Sincerely,

The Authors

Reviewer 2 Report

In this manuscript, authors have presented quad port antenna for the UWB operation. Few of the concerns are:

1. What is the application of the proposed work? It should be highlighted in the abstract section, introduction and other relevant sections. 

2. Why authors selected this frequency band for the proposed design operation? As today research is diverted towards high frequency as it is well defined that lower portion of the spectrum is highly congested by various technologies so there are chances of assigning high level of guard bands for the transmission at this particular band of interest to avoid interference with other applications. 

3. In radiation patterns comparison throughout the manuscript, it is hard to compare the simulated and measured results in E and H planes. It would be better to compare simulated and measured results within the same figure with a good visibility. 

4. In figure 8, authors have introduced strips to have polarization diversity. Do they put significant effect on decoupling level among MIMO antenna elements as well?

5. Can authors present comparison of the results with and without including strips for the MIMO antenna presented in figure 8 in terms of polarization diversity, decoupling, efficiency and gain etc.

Author Response

Dear Reviewers,

Thank you very much for your time involved in reviewing the manuscript “Compact Quad-port MIMO Antenna with Ultra-wideband and High Isolation” (ID: electronics-1977586). We also appreciate your clear and detailed feedback and hope that the explanation adequately addresses all your concerns. In the attachments, we discuss each of your comments individually, along with our corresponding responses in detail. We wish to have your approval.

To facilitate this discussion, we will first re-type your comment in italics and then state our response to the comment. Careful modifications have been made to our manuscript.

Sincerely,

The Authors

Round 2

Reviewer 1 Report

Authors have address the reviewer's comments and the manuscript has been improved.

Reviewer 2 Report

The revision seems good.